⊛ | **Open Peer Review** | Environmental Microbiology | Research Article

# Decoupling between the genetic potential and the metabolic regulation and expression in microbial organic matter cleavage across microbiomes

Zihao Zhao,[1] Federico Baltar,[1] Gerhard J. Herndl[1,2,3]

**ABSTRACT**  Metagenomics, metatranscriptomics, and metaproteomics are used to explore the microbial capability of enzyme secretion, but the links between protein-encoding genes and corresponding transcripts/proteins across ecosystems are underexplored. By conducting a multi-omics comparison focusing on key enzymes (carbohydrate-active enzymes [CAZymes] and peptidases) cleaving the main biomolecules across distinct microbiomes living in the ocean, soil, and human gut, we show that the community structure, functional diversity, and secretion mechanisms of microbial secretory CAZymes and peptidases vary drastically between microbiomes at metagenomic, metatranscriptomic, and metaproteomic levels. Such variations lead to decoupled relationships between CAZymes and peptidases from genetic potentials to protein expressions due to the different responses of key players toward organic matter sources and concentrations. Our results highlight the need for systematic analysis of the factors shaping patterns of microbial cleavage on organic matter to better link omics data to ecosystem processes.

**IMPORTANCE**  Omics tools are used to explore adaptive mechanism of microbes in diverse systems, but the advantages and limitations of different omics tools remain skeptical. Here, we reported distinct profiles in microbial secretory enzyme composition revealed by different omics methods. In general, the predicted function from metagenomic analysis decoupled from the expression of corresponding transcripts/proteins. Linking omics results to taxonomic origin, functional capability, substrate specificity, secretion preference, and enzymatic activity measurement suggested the substrate's source, concentration and stoichiometry impose strong filtering on the expression of extracellular enzymes, which may overwrite the genetic potentials. Our results present an integrated perspective on the need for multi-dimensional characterization of microbial adaptation in a changing environment.

**KEYWORDS**  multi-omics comparison, extracellular enzyme, ectoenzyme stoichiometry, secretion mechanism

O mics tools are widely used to explore the taxonomic structure and functional capability of microbial communities. The diversity, abundance, and taxonomic affiliation of genes encoding secretory enzymes have been examined by metagenomic analysis, and the corresponding regulation and expression have been investigated using metatranscriptomic and metaproteomic approaches (1–9). Yet, the links between genetic capability and transcription/protein expression remain underexplored (10). This precludes us from determining common features and distinctive characteristics among different environments inhabited by their native prokaryotic community. Extracellular enzymes are good examples to examine the connections between genetic potentials

Address correspondence to Zihao Zhao, zihao.zhao@univie.ac.at, or Gerhard J. Herndl, gerhard.herndl@univie.ac.at.

The authors declare no conflict of interest.

See the funding table on p. 14.

and metabolic regulation/synthesis because the secretion of extracellular enzymes is determined by both genetic capability and cellular response to substrate in the ambient environment (11).

Carbohydrates and proteins are the most abundant biomolecules on Earth and can be metabolized and remineralized by prokaryotes for energy and cellular metabolism (12). The size of these organic molecules like polysaccharides and proteins is too large for the prokaryotic transporter systems and consequently, not directly accessible to prokaryotes (13). Therefore, secretory enzymes initiate the first step in the hydrolysis of organic macromolecules outside of the cells, making the resulting hydrolysates available to the heterotrophic microbial community (13–15). Carbohydrate-active enzymes (CAZymes) and peptidases are the key enzymes involved in carbohydrate and protein metabolism, respectively. Enzymatic activity measurements showed that the ratio between β-glucosidase and aminopeptidase provides an indication of the balance between microbial metabolic demand and organic matter availability, and indicates the predominant lifestyle of the microbes producing those enzymes (16, 17). Secretory enzymes can occur cell-associated (in the periplasmic space or associated with the cell wall/lipopolysaccharides/S-layer) as well as in a cell-free form, hence, released into the environment. The hydrolysate produced by secretory enzymatic cleavage is transported into the cell and further metabolized by cytoplasmic enzymes either for anabolism or catabolism (18). Thus, knowledge on the characteristics (i.e., taxonomic origin, functional versatility, abundance, activity, and residence time) of prokaryotic secretory enzymes is essential to assess the role of microbes in global organic matter cleavage and cycling (19).

Hence, in this study, we investigated if there is a coupling between genetic potential and the metabolic regulation and expression in microbial organic matter cleavage across microbiomes. For that, we performed a systemic multi-environmental comparison focusing on both CAZyme and peptidase profiles using a multi-omics (metagenomics, metatranscriptomics, metaproteomics) analysis on the data from available studies. While metagenomic analyses reveal the genetic potential of microbial communities, metatranscriptomic and metaproteomic results provide complementary perspectives to the functional aspects of microbial activities. Particularly, proteins recovered from the metaproteome reflect the major functions mediated by microbes, although the resolution of metaproteomic analysis is relatively low compared to metagenomics and metatranscriptomics. Hence, multi-omics analyses provide comprehensive insights into the molecular basis of microbially mediated substrate cleavage among contrasting biomes.

The surface soil (hereafter "soil"), ocean, and human gut represent distinct habitats for microbes where the organic matter composition and availability are in sharp contrast to each other. While the major carbohydrate and protein sources for gut and soil microbes are derived from terrestrial plants and animals, marine microbes mainly thrive on algal debris and exudates, exhibiting a different C:N stoichiometry compared to terrestrial plants (marine algae, C:$N$ = 4–10:1; land plants, C:$N$ = 35–70:1) (20). The differences in C:N stoichiometry between habitats might lead to distinct genetic features in microbial communities, and the expression level of corresponding enzymes might display contrasting patterns due to the spontaneous response of the microbial community. Besides that, substrate availability for these three microbiomes represents different scenarios. In the human gut, the substrate concentration is high and easily accessible to the gut microbiota. In soil systems, however, the organic matter accessibility is limited despite its relatively high concentration. In contrast, the marine environment is characterized by low concentration of substrate and low bioreactivity, particularly in the lower mesopelagic and bathypelagic layers. It is not surprising that different habitats shape the taxonomic composition of microbes. This taxonomic change will further affect the patterns of microbial degradation of organic matter in their contrasting habitats. Thus, we hypothesize that microbial cleavage mediated by extracellular enzymes revealed by different omics tools exhibits distinct patterns between microbiomes. Furthermore, we hypothesize that these patterns will be different between the

metagenome and the -transcript/-proteome level reflecting specific adaptation patterns in response to the nutritive environment. In addition, the large size of publicly available omics data set from the ocean, soil, and human gut microbiomes enables statistical analysis and robust conclusions from the three contrasting microbial environments.

## RESULTS AND DISCUSSION

### CAZyme and peptidase profiles revealed by multi-omics analysis

In total, our multi-omics analyses were performed using microbial genes (9,878,647, 487,363,790, and 159,657,012 genes from microbiomes living in the human gut [stool sample], marine, and soil ecosystems, respectively, Table S1) (1–3, 5, 7) and publicly available metagenomes ($n = 921$), metatranscriptomes ($n = 926$), and metaproteomes ($n = 117$) (Table S2; Data Set S1) from the ocean (1–4), soil (5, 6), and human gut (7–9) to study the gene composition, transcript expression, and enzyme production of CAZymes and peptidases. In our analysis, the omics data set for gut microbiome comes from stool samples collected from citizens living in Denmark, Spain, and USA. For the marine microbiome, omics samples were collected from major ocean basins covering epi- (<200 m), meso- (200–1,000 m), and bathy-pelagic (>1,000 m) waters. For the soil microbiome, samples from surface soils (<5 cm) covering a variety of types ranging from Arctic tundra to moist tropical forests were used (Table S2; Data Set S1). This high level of heterogeneity and disparate environments might reflect environmental selection in shaping the microbial secretory enzyme pool in general, and the secretory CAZyme and peptidase pool, in particular.

By examining 656,899,449 putative genes obtained from ocean, soil, and human gut microbiomes (1–3, 5, 7), our analysis resulted in 4,816,326 CAZyme (21) and 13,409,303 peptidase (22) sequences. From those, 468,183 and 1,226,752 were predicted to encode secretory CAZymes and peptidases, respectively, as indicated by the presence of signal peptides (23). Most of the CAZyme and peptidase sequences (both secretory and cytoplasmic types) originated from bacteria in all three microbiomes (Data Set S2), while archaea contributed only marginally. The largest fraction of archaeal sequences was found in the marine microbiome, where archaeal sequences represented 1.8% and 4.2% of the secretory CAZyme and peptidase gene repertoire, respectively (Data Set S2). The contribution of marine archaea to the secretory enzyme pool was rather small as also revealed by metaproteomics (3) and extracellular enzymatic activity measurements using substrate analogs (24). Therefore, the archaeal sequences were excluded for downstream analysis.

The CAZyme and peptidase sequences were grouped into 562 and 262 functional families (see Materials and Methods), respectively. Among them, secretory CAZymes and peptidases were grouped into 499 and 237 functional families, respectively. Cytoplasmic CAZymes and peptidases covered 558 and 261 functional families, respectively. Rarefaction analysis showed a full coverage of CAZyme/peptidase families from the metagenome analysis (Fig. 1a). A large fraction of the CAZymes (88.8%, 499 out of 562) and peptidases (90.5%, 237 out of 262) contained enzymes, which can be secreted (Fig. 1a; Fig. S1a, Venn diagram). The marine and soil microbiomes shared the highest number of functional families in both CAZymes and peptidases in the secretory (Fig. 1a) as well as in the cytoplasmic (Fig. S1a) fraction. Thus, there is a higher level of shared functions among environmental microbiomes than between the environmental and the human gut microbiomes. Yet, there are also links between the gut and the environmental microbiomes as previously reported, such as the alginate lyase and carrageenase, which are expected to be present in the marine microbiome (25). However, these two enzymes are also found in the gut microbiome as a result of horizontal gene transfer from marine to gut microbes due to the traditional diet on seaweeds of Japanese (25–27).

The Shannon Index-based alpha-diversity was highest for the gut microbiome in the secretory CAZyme pool in all three omics approaches (Fig. 1b, Wilcoxon test, $P < 0.01$, Data Set S3). The marine secretory peptidase pool, however, was the most

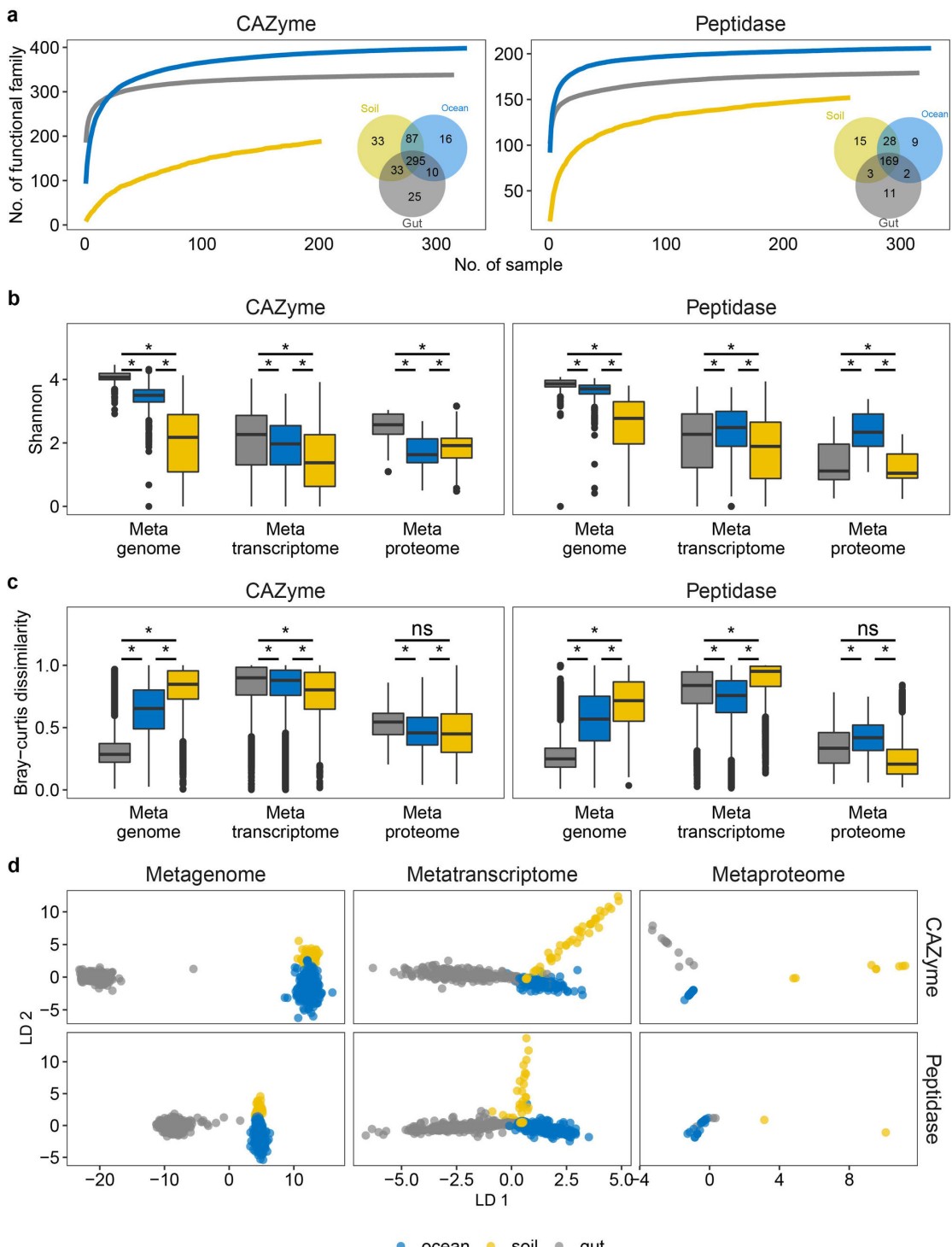

**FIG 1** Multi-omics profile of secretory CAZymes and peptidases indicates distinct distribution patterns across microbiomes. (a) Rarefaction curve of CAZyme and peptidase families. The inner Venn diagram shows the CAZyme and peptidase families identified by metagenomic assembly; numbers show the unique/shared subfamilies in each microbiome. (b and c) Shannon index-based α-diversity and Bray-Curtis dissimilarity-based β-diversity for secretory CAZymes and peptidases in multi-omics data sets. (d) Linear discriminate analysis of secretory CAZymes and peptidases in multi-omics data sets. Statistics are based on Wilcoxon test (*, $P < 0.05$; ns, not significant).

diverse at the metatranscriptomic and metaproteomic levels, while the gut microbiome exhibited the highest diversity in the metagenome (Fig. 1b, Wilcoxon test, $P < 0.01$,

Data Set S3). Permutational multivariate analysis of variance (PERMANOVA) showed that the functional composition of CAZymes and peptidases was different between microbiomes in all three omics data sets (*Adonis* function in R, $P < 0.05$). The Bray-Curtis dissimilarity-based beta-diversity revealed that environmental microbiomes (i.e., ocean and soil) exhibit a higher (Wilcoxon test, $P < 0.01$, Data Set S3) variance than the human gut microbiome at the metagenomic level in both the secretory CAZyme and peptidase pool (Fig. 1c). In the metatranscriptome and metaproteome, however, the beta-diversity was similar between microbiomes, especially for the CAZyme pool. The linear discriminant analysis (28) also showed habitat-based separation patterns at all three omics data sets for both CAZymes and peptidases (Fig. 1d), which might be caused by differences in organic matter availability and/or physicochemical parameters between the microbiomes (i.e., aerobic vs anaerobic, acidic vs alkaline, diffusion-limited vs diffusive, high vs low salt concentration).

We further examined the clustering pattern of genes encoding secretory CAZymes and peptidases for each microbiome. In the ocean, secretory CAZyme/peptidase genes clustered according to depth, and that of the human gut microbiome clustered by nation (Fig. S2). The depth-driven clustering pattern in the marine environment is consistent with a decline in substrate concentration and quality in the water column with depth (29). The clustering pattern of the human gut microbiome from citizens living in different nations is probably caused by differences in food sources and cooking style (30). In contrast, the clustering pattern of the soil microbiome was not that clear across different soil types except for the microbes inhabiting boreal forests, the latter representing nutrient-rich conditions (5). These results further support the hypothesis that the changes in substrate composition and availability affect the microbial CAZymes and peptidases globally as well as regionally.

Cytoplasmic CAZymes and peptidases exhibited a pattern similar to their secretory counterparts (Fig. S1b through d), indicating a tight link between the cellular metabolism and the extracellular and/or surface-associated organic matter cleavage and acquisition.

## Taxonomic and functional heterogeneity of CAZymes and peptidases

To further elucidate how environmental heterogeneity affects bacterial utilization of carbohydrates and proteins, we investigated the association between the functional capacity and the taxonomic composition of CAZymes and peptidases across the three microbiomes. We found fundamental differences between the microbiomes at the three omics levels for both the functional capacity and the taxonomic composition (Fig. 2; Fig. S3). In the secretory CAZyme pool of the human gut (Fig. 3a and b), Bacteroidetes (77.1 ± 13.7% in the metagenome, 57.3 ± 30.6% in the metatranscriptome, and 45 ± 13.8% in the metaproteome, Data Set S4) and Firmicutes (9.6 ± 9.5% in the metagenome, 48.3 ± 35.1% in the metatranscriptome, and 36.8 ± 19.6% in the metaproteome, Data Set S4) were the main producers of CAZymes. Both of them can secrete glycoside hydrolases (GHs, 84.2 ± 2.5% in metagenome, 79.5 ± 17.5% in metatranscriptome, and 83.3 ± 8.5% in metaproteome, Data Set S5). GHs were also abundant in soil (55.2 ± 30.4% in the metagenome, 61.7 ± 30.5% in the metatranscriptome, and 73.7 ± 14.7% in the metaproteome, Data Set S5) and marine (54.4 ± 11.6% in the metagenome, 47.1 ± 28.6% in the metatranscriptome, and 28.5 ± 20.4% in the metaproteome, Data Set S5) microbiomes. Polysaccharide lyases (PLs) were more abundant in the human gut and marine microbiomes than in soils; whereas, carbohydrate esterases (CEs) were enriched in the soil and marine environments as compared to the human gut (Fig. 2a, Wilcoxon test, $P < 0.01$, Data Set S5). CAZymes with auxiliary activities (AAs) are oxidative enzymes and only present in the marine and soil microbiomes. In the marine microbiome, although genes encoding AAs only constituted 8 ± 4.7% of the total CAZyme gene pool, their expression level was high in the metatranscriptome and metaproteome (37.3 ± 31.8% in the metatranscriptome and 51.1 ± 21% in the metaproteome, Data Set S5). CAZymes with AAs are known as lytic polysaccharide monooxygenases (LPMOs) (31) and are involved in polysaccharide degradation because crystalline polysaccharides in higher

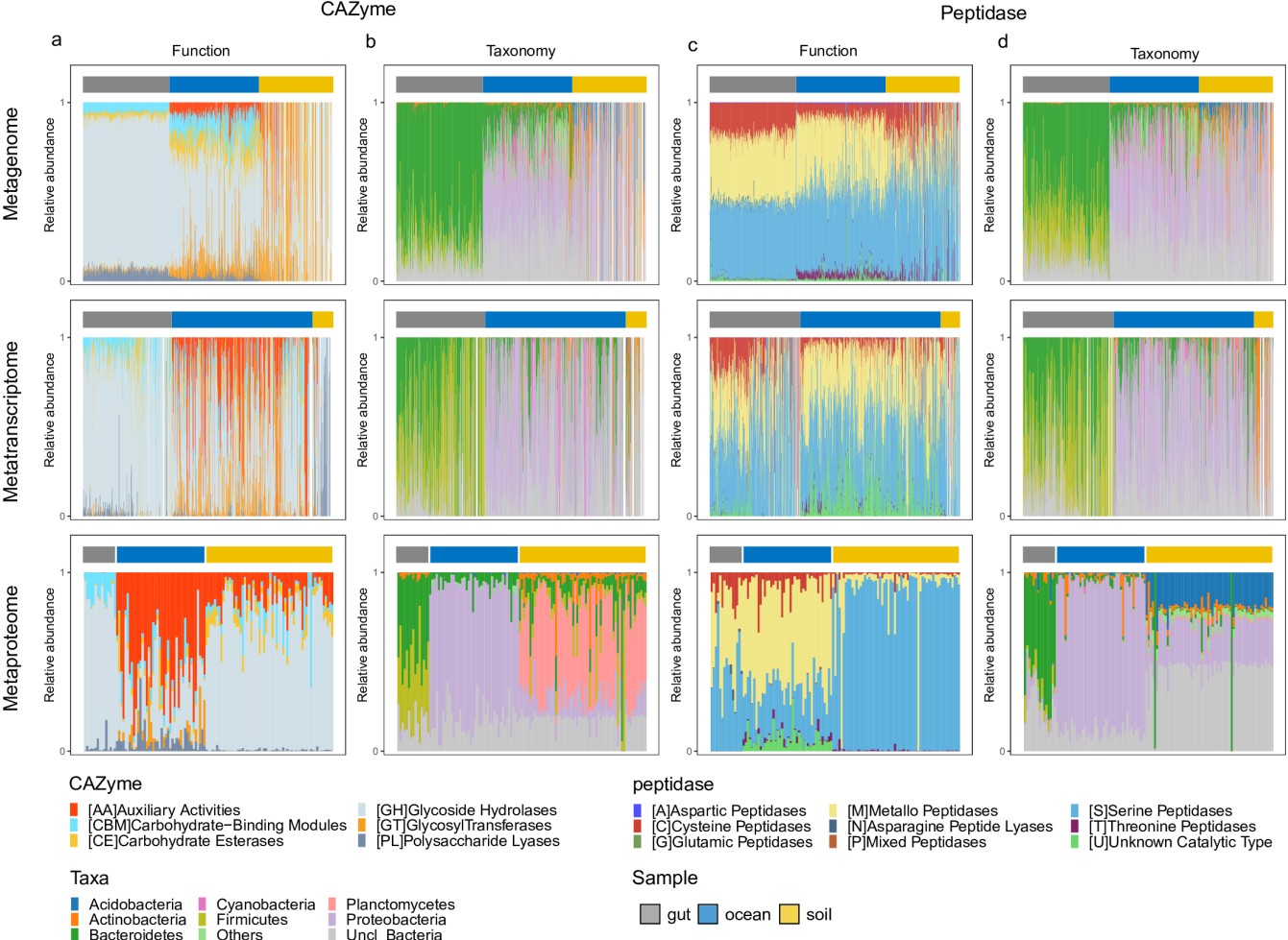

**FIG 2** Taxonomic and functional variation of secretory CAZymes and peptidases in multi-omics data sets. (a and b) Taxonomic and functional variation of secretory CAZymes in meta-genomic, -transcriptomic, and -proteomic data sets. (c and d) Taxonomic and functional variation of secretory peptidases in meta-genomic, -transcriptomic, and -proteomic data sets.

plants and algae are less accessible to degradative CAZymes (like GHs, CEs, and PLs) (32, 33). By oxidizing the surface of polysaccharides' crystal structure and disrupting the polysaccharides' topology, the oxidative activity of CAZymes with AAs creates tractable chain ends for further depolymerization mediated by hydrolytic CAZymes like GHs (34).

Yet, the taxonomic composition of secretory CAZymes differed drastically between the soil and the marine microbiomes in all three omics data sets (Fig. 2b). Bacteroidetes (16.9 ± 14.6% in the metagenome, 20.6 ± 24% in the metatranscriptome, and 6.2 ± 5.4% in the metaproteome, mainly Flavobacterales, Data Sets S4 and S6) and Proteobacteria (36 ± 18.7% in the metagenome, 58.4 ± 33.6% in the metatranscriptome, and 75.3 ± 10% in the metaproteome, especially Alphaproteobacteria and Gammaproteobacteria, Fig. S3; Data Sets S4 and S6) were the main secretory CAZyme producers in the ocean, in agreement with a previous report (3). However, Planctomycetes were identified by metaproteomic analysis (45.6 ± 21.5%) as the major CAZyme producers in soil, which is in contrast to their low contribution to CAZymes in the metagenome and metatranscriptome (representing only 3.6 ± 3.6% and 4.1 ± 4.4%, respectively, Data Set S4). Fluorescent *in situ* hybridization-based studies show that Planctomycetes account for 4% to 7% of total cells in agricultural and forest soils, and 16S rRNA gene studies report a relative contribution of 7 ± 5% to the total bacterial community in agricultural soils (35–37). These observations are consistent with our results predicted from the metagenomic data

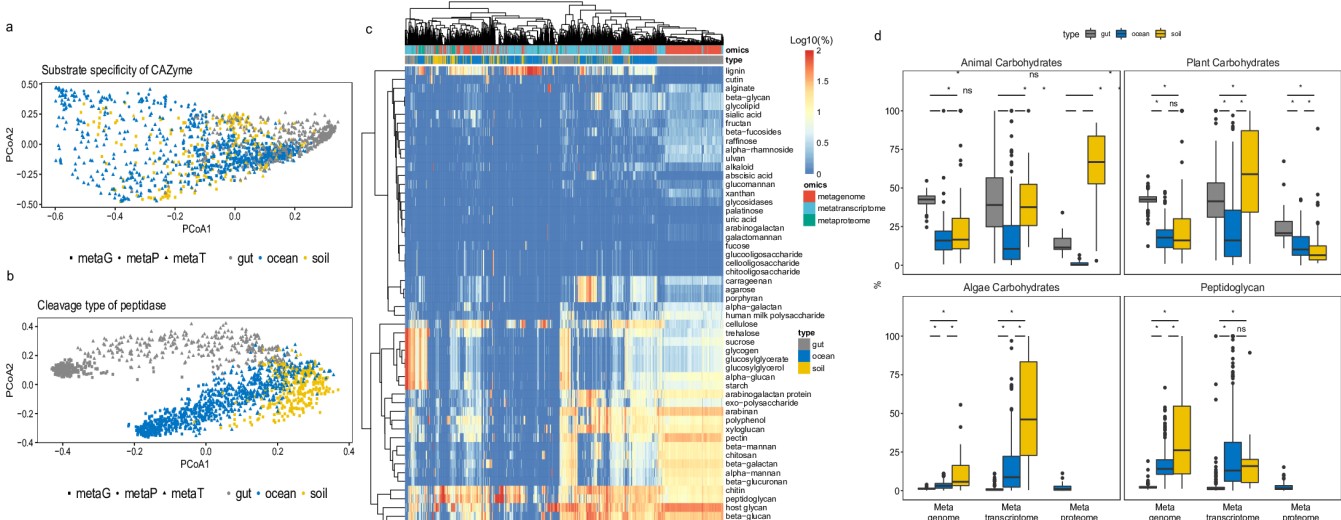

**FIG 3** Cleavage profile of secretory CAZyme and peptidase. Substrate specificity of CAZymes (a) and cleavage type of peptidases (b) show distinct clustering pattern at multi-omics level between microbiomes. (c) Heat map of CAZyme substrate specificity in microbiomes living in the ocean, soil, and human gut revealed by omics data set. (d) Relative abundance of substrate-specific CAZymes in multi-omics data sets. Statistics are based on Wilcoxon test (*, $P < 0.05$; ns, not significant).

set where Planctomycetes-affiliated CAZyme genes accounted for only 3.6 ± 3.6% of the secretory CAZyme gene pool in soils (Fig. 2b). This low representation of Planctomycetes is in striking contrast to their major role in secretory CAZyme production in the soil metaproteome (Fig. 2b; Data Set S4). In the soil, Planctomycetes-affiliated secretory CAZymes contributed 45.6 ± 21.5% to the secretory CAZyme pool and and their relative abundance was significantly higher than that of the corresponding genes (Wilcoxon test, $P < 0.05$). This also represents an example of the importance of studying not only the gene potential via metagenomic analyses but also the protein expression, to elucidate the actual role of microbes in the environment. Such capacity in organic carbon cycling might explain the key role of Planctomycetes in soil recovery from the adverse effects of the extensive use of fertilizers in agriculture (38). Besides Planctomycetes, other major producers of secretory CAZymes in soils are Actinobacteria, Bacteroidetes, and Proteobacteria. Alpha- and Gamma-proteobacteria in the soils mainly secreted CEs as degradative CAZymes, which is in sharp contrast with marine phylotypes using GHs and PLs for carbohydrate hydrolysis (Fig. S4 and S5). It may be argued that the size of soil metaproteomic samples is much lower than the metagenomic samples; however, those soil metaproteomes are collected from soils with a wide range of carbon contents total organic carbon [(TOC) range: 0.26%–15%, dissolved organic carbon (DOC) range: 20.7–1,596 mg C kg$^{-1}$]. Hence, the dominance of Planctomycetes in secretory CAZyme production provides strong indication for the uncoupling between genetic potential and microbial metabolic activity.

The functional composition of the secretory peptidases differed from that of the secretory CAZyme pool. The marine microbiome exhibited high functional similarity with the gut microbiome, with a preference for cysteine (C)-, metallo (M)-, and serine (S) peptidases (Fig. 2c). This is in contrast to the soil microbiome, which mainly produced serine (S) peptidase (88.2 ± 19.4% in the metaproteome, Data Set S5). For the marine and gut microbiomes, the taxonomic composition of secretory peptidases in the three omics data sets was similar to that of the secretory CAZyme pool (Fig. 2b and d). Hence, the major secretory CAZyme producers also secreted peptidases. In the soil microbiome, however, Acidobacteria and Proteobacteria (besides unclassified bacteria) were the main producers of secretory peptidases (Fig. 2b and d), which is consistent with their reported abundance (39). Planctomycetes-affiliated peptidases were barely

detected (4.2 ± 5.4% in the metagenome, 3.4 ± 1.3% in the metatranscriptome, and 0.9 ± 0.3% in the metaproteome, Data Set S4), indicating that in the soil microbiome, Planctomycetes target mainly organic carbon from plant carbohydrates and Acidobacteria and Proteobacteria nitrogenous organic matter from animals. A similar scenario has been reported for the deep-sea environment, where Gammaproteobacteria secrete both CAZymes and peptidases, while Alphaproteobacteria show a preference for secretory peptidase production (3). In addition, the functional composition of marine Proteobacteria (mainly Alphaproteobacteria and Gammaproteobacteria) also differed from that of soil Proteobacteria. Marine Proteobacteria preferentially secreted cysteine (C)- and metallo (M)-peptidases; however, soil Proteobacteria used serine (S)- and threonine (T)-peptidases (Fig. S4 and S5). Recent findings indicate that marine Gammaproteobacteria can use cell-free secretory peptidases (metallo-peptidase) to outcompete Gram-positive bacteria for substrate (40). In addition, secretory peptidases with unknown catalytic activities (U) were relatively abundant in the marine metatranscriptomic and metaproteomic data set (Fig. 2c; Fig. S4), which might be responsible for the high diversity in the marine secretory peptidase pool (Fig. 1b). These results suggest a more diverse organic nitrogen source for marine than for soil Proteobacteria. Firmicutes in the human gut exhibited a low secretory capacity for peptidases (6.2 ± 4.1% in the metaproteome) despite their high potential at the gene (23.9 ± 15.1% in the metagenome) and transcription levels (41.7 ± 30.4% in the metatranscriptome) as indicated also by the low functional diversity in secretory peptidases in the human gut at the metaproteome level (Fig. 1b).

There were also significant differences between the secretory and cytoplasmic enzyme pools for both CAZymes and peptidases in all three microbiomes (Fig. 2; Fig. S3 to S6). The glycosyl transferases (GTs), which were rarely found in the secretory CAZyme pool (Fig. 2a; Fig. S6a), constituted the major fraction of the cytoplasmic CAZyme pool in all three microbiomes (Fig. S3a). Also, the threonine peptidases (T) found in the soil cytoplasmic peptidase pool (Fig. S3c and S6b) were absent in the secretory peptidase pool in the soil microbiome (Fig. 2c). The proportions of cysteine peptidase (C) were also lower in the secretory than in the cytoplasmic peptidase pool especially in the marine microbiome (Fig. 2c; Fig. S3c and S6b). Such differences between the secretory and cytoplasmic CAZymes/peptidases in all three microbiomes are probably caused by different metabolic requirements. For example, GTs are mainly involved in the formation of the glycosidic linkage to form a glycoside, which occurs in the cytoplasm (41). The differences between the secretory and cytoplasmic pool resulted in large variations in the overall capacity of enzyme secretion between microbiomes (Fig. S7). The gut microbiome has a high capacity to secret CAZymes like CEs, GHs, and PLs, and peptidases like cysteine (C)-, metallo (M)-, and asparagine peptide lyase (N-). The marine and soil microbiomes, however, secret CAZymes with auxiliary activities and serine peptidase for extracellular hydrolytic activities. The taxonomic affiliation of the secretory and cytoplasmic fraction also revealed that five phyla (i.e., Actinobacteria, Acidobacteria, Bacteroidetes, Gammaproteobacteria, and Planctomycetes; Fig. 2, Fig. S3) are the major producers of extracellular enzymes, while their secretion capacity varied among microbiomes (Fig. S8).

One major difference between these three microbiomes is that the sources of carbohydrates and proteins are different (terrestrial plants vs marine algae, terrestrial animals vs marine zooplankton). To examine how substrate composition affects the microbially mediated substrate cleavage, we looked into the substrate specificity of CAZymes and the cleavage type of peptidase among ocean, soil, and human gut microbiomes as revealed by the different omics approaches (Fig. 3a and b). A significant clustering pattern shaped by both omics tools and microbiome types was found for both CAZyme substrate specificity (type and omics: degrees of freedom (Df) = 4, Sum of sum of squares (Sqs) = 30.84, $R^2$ = 0.07, F = 46.433, P < 0.001; Fig. 3a, Fig. S8) and peptidase cleavage types (type and omics: Df = 4, Sum of Sqs = 44.79, $R^2$ = 0.06, F = 44.084, P < 0.001; Fig. 3b, Fig. S9). We specifically investigated the CAZymes (Fig.

3c and d; Table S3) targeting carbohydrates of different origin (42). We found that both gut and soil microbiomes exhibited a high capability to degrade carbohydrates originating from animals and plants, not only at the gene level but also at the protein level. In contrast, CAZymes degrading algal carbohydrates were only found in the marine metaproteome, although the genes and transcripts of those CAZymes were also found in the gut and soil microbiomes. This high level of gene/transcript abundance of CAZymes targeting algal carbohydrates in soil microbiome is because the GH55 and GH16 are also widely used by soil microbes hydrolyzing glycosidic bonds in various glucans, and laminarin and carrageenan in marine environment are also rich in β-1,4 or β-1,3 glyosidic bonds. Similarly, genes/transcripts encoding CAZymes to hydrolyze peptidoglycans (bacterial cell wall) were widespread among the three microbiomes but their expression was only detected in the marine metaproteome. Peptidoglycan is an essential component of the bacterial cell wall and can be released during cell death. Besides CAZymes, metallo-peptidase (M23) is also required to degrade peptidoglycan. We further explored the profile of M23 in all three microbiomes and found that M23 was primarily produced by marine microbes in the secretory peptidase pool (Fig. S10). Hence, together with the CAZyme profile, marine microbes exhibit a strong tendency to recycle peptidoglycan together with cellular components released upon cell decay (43). It should be mentioned, however, that the resolution of metaproteomics is lower than that of metagenomics and metatranscriptomics. Thus, the expression of CAZymes and peptidases targeting peptidoglycan might have been overlooked in the metaproteome of soil and gut microbiomes due to low expression levels, because peptidoglycan needs to be broken down during cell division. However, it can be argued that marine microbes exhibit the lower growth rates than gut and soil microbes (44), but the expression for enzymes targeting peptidoglycan degradation is most prominent in the marine metaproteome. Hence, it is likely that marine bacteria are better adapted to recycle bacterial debris than gut and soil bacteria.

## Secretion mechanisms for CAZymes and peptidases vary among microbiomes

To understand the secretion mechanisms used by bacteria in the contrasting environments, we examined the types of signal peptides for secretory CAZymes and peptidases, as well as the possible secretion pathways (Fig. 4a; Data Set S7). In the gut microbiome, 65% of the secretory CAZymes contained type I signal peptides (SPI) and 25% of the peptidases contained type II signal peptides (SPII), and both were secreted by the general secretion pathway (Sec)" to refer to CAZymes and peptidase. Only <5% of secretory CAZymes and peptidases with SPI were secreted by the twin-arginine translocation pathway (TAT). In contrast, in the marine environment, a higher fraction of secretory CAZymes and peptidases contained the SPII and TAT system than in the gut microbiome (Fig. 4a). The TAT system is also widely used by the soil microbiome as revealed by metaproteomic analysis showing that about 50% of the secretory CAZymes and 70% of the secretory peptidases are exported by the TAT system (Fig. 4a). This suggests that while SPI is the general signal peptide used by all microorganisms, there are major differences in the export mechanism for SPI-containing secretory CAZymes and peptidases across microbiomes. In contrast to the human gut microbiome, marine and particularly soil bacteria show a preference to use the TAT system for exporting secretory CAZymes and peptidases. Despite the high energy investment and low transportation speed, the benefit of the TAT system is that enzymes like CAZymes and peptidases are exported after folding, which takes place in the cytoplasmic space and leads to enzyme maturity (45). Because protein folding is critical for enzymatic activity and is largely affected by parameters like pH, temperature, and salt concentrations (46), the TAT system enables secretory enzymes to reach their biologically active state prior to exportation (45).

We also investigated the link between the secretion mechanism and enzyme properties by predicting the isoelectric points (pIs) of putative CAZymes and peptidases.

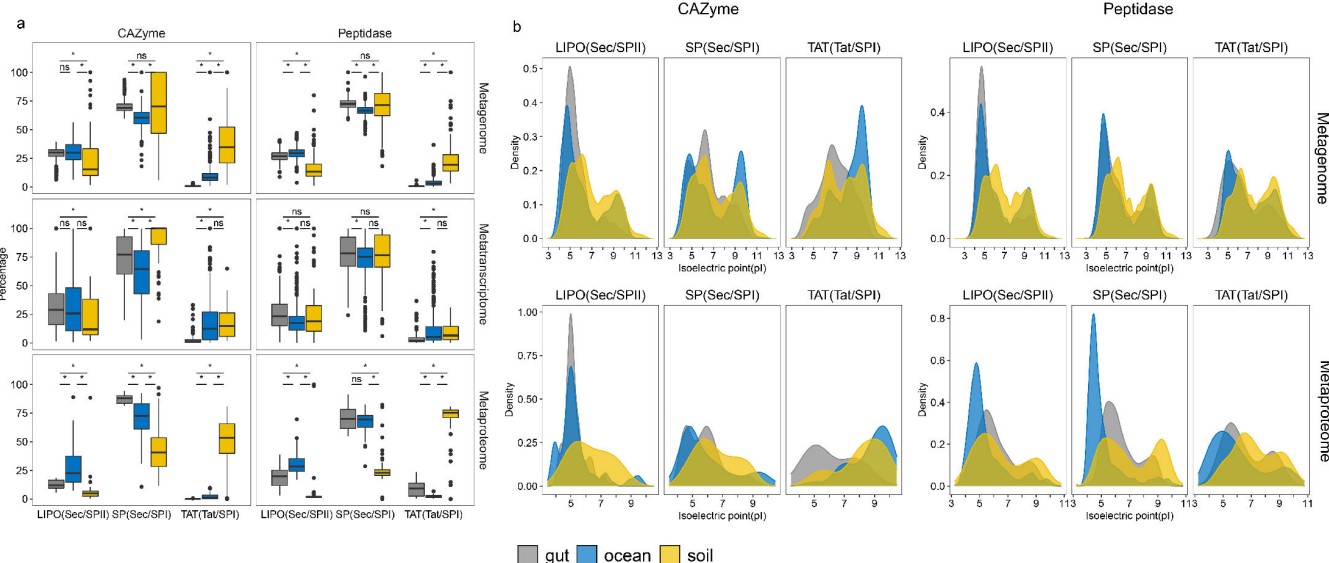

**FIG 4** Predicted secretion mechanisms and protein pI of secretory CAZymes and peptidases of the ocean, soil, and human gut microbiomes. (a) Percentage of secretion pathway in multi-omics data sets. (b) pI distribution of putative (upper) and identified (lower) secretory CAZymes and peptidases exported by different secretion pathways. SP(Sec/SPI): genes encoding secretory enzymes with type I signal peptide exported through general secretion pathway; TAT(TAT/SPI): genes encoding secretory enzymes with type I signal peptide exported through twin-arginine translocation pathway; LIPO(Sec/SPI): genes encoding secretory enzymes with type II signal peptide exported through general secretion pathway. Statistics are based on Wilcoxon test (*, $P < 0.05$; ns, not significant).

The pI determines the solubility (especially for cell-free enzymes) and catalytic performance of the enzyme (47). A bimodal distribution pattern of pI was found for both cytoplasmic and secretory CAZymes and peptidases at the metagenome and metaproteome levels (Fig. S11). According to the pI analysis, the CAZymes and peptidases were grouped into acidic and alkaline types. Acidic types were more prevalently encoded in the human gut and soil microbiomes, although they were also present in the marine microbiome (Fig. S11). Marine bacteria encoded more alkaline CAZymes than the gut and soil microbiomes, which was reflected in the metaproteome (Fig. 4b). Both marine and soil microbes use the TAT system to secret alkaline CAZymes. The use of the TAT system for peptidase secretion, however, was largely limited to soils (Fig. 4b). The pI of enzymes is related to their stability (48). Thus, alkaline enzymes exhibit a longer half-life time than acidic ones (48). Also, the protein becomes aggregated under a pH close to its pI (47). Therefore, alkaline enzymes show higher stability against aggregation. In addition, alkaline CAZymes with a longer half-life time or higher stability might lower the costs for cell-free secretory enzyme production and maintenance of the cell-free secretory CAZyme pool in the environment.

## Substrate stoichiometry shapes enzyme expression

To further decipher how omics data are associated with the bacterial utilization of organic matter, we investigated the connection between omics data to enzymatic stoichiometry. β-glucosidase (BG) and leucine aminopeptidase (LAP) are widely used in marine and soil environments to examine microbial cleavage capabilities on carbohydrates and proteins (16, 49), and the ratio between BG and LAP has been shown to reflect the nutrient availability across environments (16). Here, we first examine the relationship between secretory CAZymes and peptidases on omics levels. We analyzed the abundance of secretory CAZymes (with degradative functions: i.e., CEs, GHs, and PLs) and peptidases, as well as the proportion of secretory to the total CAZyme and peptidase pool (Fig. 5). A strong positive correlation was found between genes encoding secretory CAZymes and peptidases (and the proportion of secretory to total) across all three microbiomes, both for the whole community (Fig. 5a and d) and for specific taxa

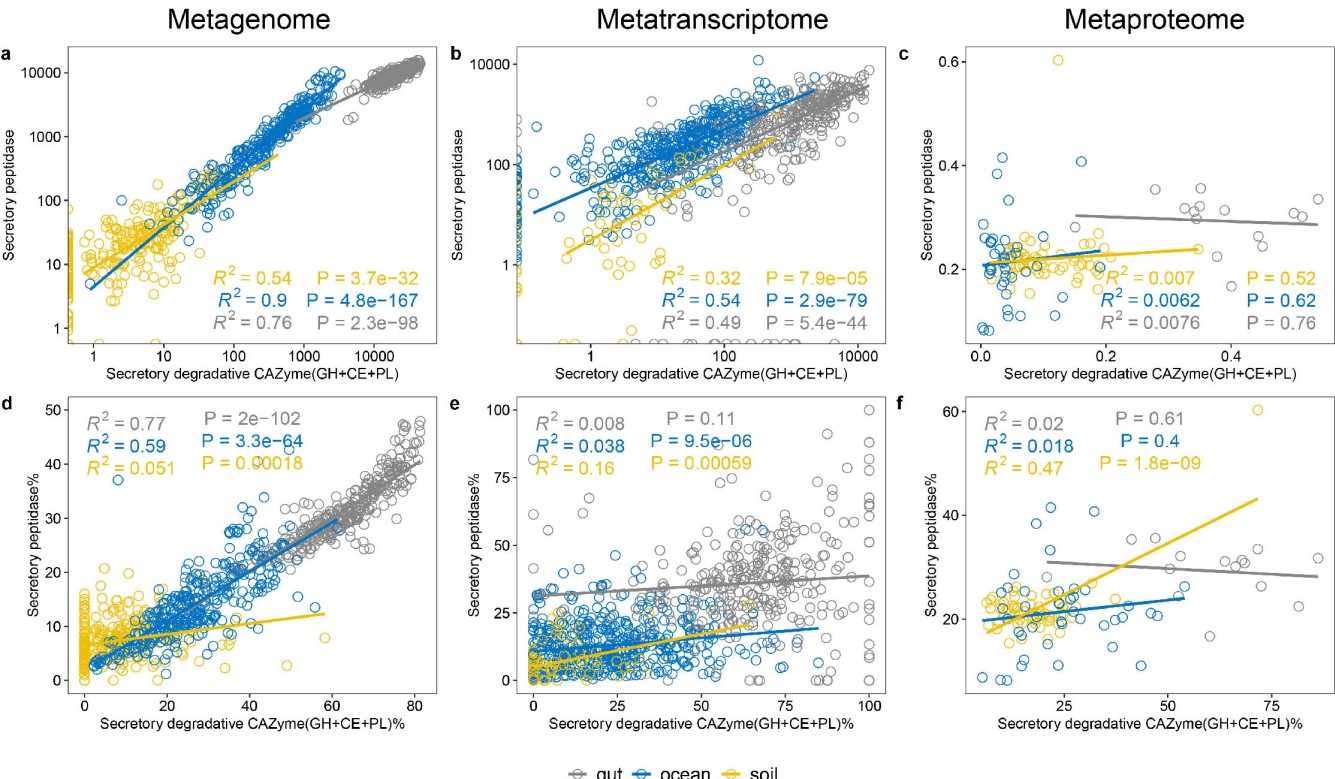

**FIG 5** Relation between secretory degradative CAZymes and secretory peptidases in multi-omics data sets. (a–c) While the genes and transcripts of secretory degradative CAZymes are related to that of secretory peptidases in the metagenome and metatranscriptome. (d–f) The proportion of secretory degradative CAZymes to total (secretory and cytoplasmic) degradative CAZymes is also related to the proportion of secretory peptidases to total peptidases in the metagenome, but such relationship is not found in the metatranscriptome and metaproteome.

(Fig. S12 and S13). However, these correlations found in the metagenomes were weak or not existing in the metatranscriptome and metaproteome levels (Fig. 5b, c, e, and f,; Fig. S12 and S13). A similar trend was also observed for cytoplasmic CAZymes and peptidases (Fig. S14). This decoupling between the genetic potential and the metabolic regulation and expression might reflect intensive environmental selection (i.e., bacterial interactions with the surrounding environment as well as within the community) on organic matter cleavage. Consistent with the functional and taxonomic analyses of CAZymes and proteases (Fig. 2; Fig. S4), the differences observed between metagenomes and metaproteomes suggest that specific functions of certain taxa (i.e., Bacteroidetes in human gut, Gammaproteobacteria in the ocean, and Planctomycetes in the soil) dominate the ectoenzyme pool as a result of their metabolic response/adaption to their particular environment. This, in turn, indicates that the characteristics and availability of organic matter affect the expression and exportation of secretory enzymes (50), explaining the decoupling between secretory CAZymes and peptidases at the metagenomic and metaproteomic levels across ecosystems.

## Conclusion

Despite the potential limitations associated with meta-omics analyses (such as sampling size, lack of paralleled multi-omics data sets, etc.), our comparative analysis provides a systematic insight into the distribution of bacteria ectoenzymes and their expression to cleave polysaccharides and proteins across microbiomes (i.e., marine, soil, human gut). Our results reveal that the key taxonomic microbial groups in organic matter hydrolysis living in different habitats can be differentiated by omics tool, and the changes in enzyme production between different taxa lead to a decoupling between genomic

potential and corresponding transcripts/proteins. Key players expressing a larger fraction of the enzymes than expected based on their genomic potential cause this uncoupling between genetic potential and enzyme expression. These key players probably act as the driving force in the element cycling. Thus, studies based exclusively on metagenomic analyses to elucidate functions in microbial communities do not reflect the actual distribution patterns of ectoenzyme secretion. We also found evidence that organic matter properties regulate the microbial secretion of enzymes for hydrolyzing macromolecules. Bacteria adjust either the characteristics of their enzymes (e.g., the pI) or the secretion mechanism (e.g., secreting through the TAT system to lower the risk of failure in secretory enzyme production). Taken together, integrated analysis using multi-omics tools may allow better interpretation on the molecular basis of microbially mediated biogeochemical cycling.

## MATERIALS AND METHODS

### CAZyme and peptidase gene catalog construction

Predicted genes (both nucleic acid and amino acid sequences) from metagenomic assemblies were downloaded from the databases (1, 3, 5, 7). Sequences encoding CAZymes were annotated using hmmsearch against the dbCAN database (dbCAN-HMMdb-V8) (51) with the threshold of domain coverage >35% and an e-value $<1 \times 10^{-18}$. Peptidases were annotated with Diamond (0.8.36) (52) blast against the MEROPS database (downloaded in March 2023) (22) using cutoffs of e-value $<1 \times 10^{-20}$ (% of identity >30%). Functional annotation of CAZyme/peptidase sequences was carried out by assigning sequences into functional families (CAZyme families: auxiliary activities [AA], carbohydrate-binding modules [CBM], carbohydrate esterases [CE], glycoside hydrolases [GH], glycosyl transferases [GT], and polysaccharide lyases [PL]; peptidase families: aspartic peptidases [A], cysteine peptidases [C], glutamic peptidases [G], metallo peptidases [M], asparagine peptide lyases [N], mixed peptidases [P], serine peptidases [S], threonine peptidases [T], and unknown catalytic type [U]). Subfamilies were treated as families, for example, GH16_1 and GH16_2 were treated as two families instead of grouping them into the GH16 family. Peptidases were treated similarly. The taxonomic affiliation of corresponding sequences was identified using the lowest common ancestor algorithm adapted from DIAMOND (0.8.36) (52) by searching against the NCBI non-redundant (NR) database. The top 10% hits with an e-value $<1 \times 10^{-5}$ were used for taxonomic assessment (--top 10). Only prokaryotic sequences (archaea and bacteria) were retained. Seqkit (53) toolkit (seqkit rmdup) was used to remove identical sequences (similarity 100%) and then followed by a further de-redundancy process with CD-HIT (4.6.8) (54). General substrate prediction of CAZymes was derived from prediction files in dbCAN database (https://bcb.unl.edu/dbCAN2/download/Databases/fam-substrate-mapping-08252022.tsv). Substrate specificity of CAZymes targeting carbohydrates originating from animals, plants, and bacteria was summarized from a previous report (42). CAZymes targeting algal carbohydrates were determined by searching key words "laminarin," "carrageenan," "ulvan," "fucoidan," "alginate," and "agarose" in CAZypedia (https://www.cazypedia.org/index.php/CAZypedia). The detailed classification is listed in Table S3. To evaluate the relative abundance of CAZymes targeting specific carbohydrates in the secretory CAZyme pool in the omics data sets, the gene/transcript/protein abundance of CAZymes affiliated to families listed in Table S3 was summed and normalized to the abundance of all secretory CAZymes.

### Signal peptide, secretion mechanism, and pI prediction

SignalP (5.0) (23) was used to detect the presence of signal peptides for prokaryotic sequences as well as possible secretion pathways. Archaeal sequences were predicted under archaeal mode (-arch). The sequences of Actinobacteria- and Firmicutes-affiliated sequences were predicted under Gram-positive mode (-gram+), while other

bacterial sequences were predicted under Gram-negative mode (-gram-). The pI for each sequence was predicted with protein isoelectric point calculator (IPC) (55). The average pI value from 17 prediction models (Bjellqvist, DTASelect, Dawson, EMBOSS, Grimsley, IPC_peptide, IPC_protein, Lehninger, Nozaki, Patrickios, ProMoST, Rodwell, Sillero, Solomon, Thurlkill, Toseland, Wikipedia) was taken for analysis.

## Recruitment of metagenomic and metatranscriptomic reads

Metagenomic and metatranscriptomic reads were downloaded from NCBI according to the accession numbers provided in the respective publications (Data Set S1). The marine metagenomes/metatranscriptomes were collected from global ocean expeditions (Tara Ocean, Malaspina, and Geotraces) covering the entire water column from 5 to 4,000 m (1–4). The soil metagenomes/metatranscriptomes cover microbiomes of 11 soil types ranging from boreal forests to the polar tundra (5, 6). Metagenomes/metatranscriptomes from gut microbiomes of citizens of Denmark, Spain, and USA were used for the analyses (7–9). Reads from each metagenome/metatranscriptome were mapped to the CAZyme/peptidase gene catalog with Burrows-Wheeler Aligner (BWA) (0.7.16a) (56) algorithm. Gene abundance was estimated from sequences with >95% read coverage and was normalized as follows: gene abundance = $10^6 \times$ (mapped reads/gene length)/total reads.

## Metaproteomic profiling and analysis

The CAZyme and peptidase sequences identified from genes predicted from metagenomics assembles were clustered at 90% similarity (-c 0.9 -G 0 -aS 0.9) using CD-HIT (4.6.8) (54) to establish a curated database for metaproteomic profiling. The tandem mass spectrometry spectra from each proteomic sample were downloaded from Proteome Identifications (PRIDE) Database and were searched using SEQUEST-HT (57). Mapped spectra were further validated with the Percolator in Proteome Discoverer 2.1 (Thermo Fisher Scientific). False discovery rate in the target-decoy approach (58) was set to 1% at the peptide level as the threshold to reduce the probability of false positive identification. A minimum of two peptides and one unique peptide (peptide only mapped to one gene in the database) were required for protein identification. Quantification of the relative abundance of proteins was conducted with a chromatographic peak area-based, label-free quantitative method (59), where the peak areas of unique peptides were summed and normalized to the normalized area abundance factor (NAAF).

## Statistical analysis and visualization

Statistics and visualization were performed in R (https://www.r-project.org/). *Vegan*, *rtk*, and *wilcox.test* were used for ordination, diversity calculation, and significance test; *ggplot2* was used for data visualization.

### ACKNOWLEDGMENTS

The study was supported by the Austrian Science Fund (FWF) projects ARTEMIS (P28781-B21 to G.J.H.) and DECOMB (I 4978-B to G.J.H.).

We thank the reviewers for their insightful comments, which helped to improve the quality of the manuscript.

Z.Z. and G.J.H. conceived the project. Z.Z. collected the metagenomic data sets and performed the analysis. Z.Z., F.B., and G.J.H. interpreted the data and wrote the paper.

### AUTHOR AFFILIATIONS

[1]Department of Functional and Evolutionary Ecology, Bio-Oceanography Unit, University of Vienna, Vienna, Austria

[2]NIOZ, Department of Marine Microbiology and Biogeochemistry, Royal Netherlands Institute for Sea Research, Den Burg, the Netherlands

[3]Vienna Metabolomics Center, University of Vienna, Vienna, Austria

## AUTHOR ORCIDs

Zihao Zhao http://orcid.org/0000-0001-7497-3276
Gerhard J. Herndl http://orcid.org/0000-0002-2223-2852

## FUNDING

| Funder | Grant(s) | Author(s) |
| --- | --- | --- |
| Austrian Science Fund (FWF) | P28781-B21, I4978-B | Gerhard J. Herndl |

## AUTHOR CONTRIBUTIONS

Zihao Zhao, Conceptualization, Data curation, Formal analysis, Investigation, Methodology, Software, Visualization, Writing – original draft, Writing – review and editing | Federico Baltar, Data curation, Methodology, Writing – review and editing | Gerhard J. Herndl, Project administration, Supervision, Writing – review and editing

## DATA AVAILABILITY

Prokaryotic sequences from metagenomic assemblies and metagenomic reads analyzed in this work are publicly available. Corresponding references and accession numbers can be found in the Data Set S1.

## ADDITIONAL FILES

The following material is available online.

### Supplemental Material

**Data Set S1 (Spectrum03036-23-s0001.xlsx).** Accession numbers for omics data set.
**Data Set S2 (Spectrum03036-23-s0002.xlsx).** CAZyme/peptidase sequences distribution between archaea and bacteria.
**Data Set S3 (Spectrum03036-23-s0003.xlsx).** Wilcoxon test result for diversity index.
**Data Set S4 (Spectrum03036-23-s0004.xlsx).** Relative abundance of taxonomic groups contributing to the CAZyme/peptidase pool in omics data set.
**Data Set S5 (Spectrum03036-23-s0005.xlsx).** Functional composition of the CAZyme/peptidase in omics data set.
**Data Set S6 (Spectrum03036-23-s0006.xlsx).** Relative abundance of major taxa (class level) contributing to the CAZyme/peptidase pool in omics data set.
**Data Set S7 (Spectrum03036-23-s0007.xlsx).** Wilcoxon test result for CAZyme/peptidase secretion pathway employment.
**Supplemental figures and tables (Spectrum03036-23-s0008.pdf).** Figure S1 to S14; Table S1 to S3.

### Open Peer Review

**PEER REVIEW HISTORY (review-history.pdf).** An accounting of the reviewer comments and feedback.

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
