## [Reviewer comments · Microbiology Spectrum]

Microbiology Spectrum

Decoupling between genetic potential and the metabolic regulation and expression in microbial organic matter cleavage across microbiomes

Zihao Zhao, Federico Baltar, and Gerhard Herndl

Corresponding Author(s): Zihao Zhao, Universitat Wien

Review Timeline:

Submission Date:	August 4, 2023
Editorial Decision:	December 22, 2023
Revision Received:	February 7, 2024
Accepted:	March 6, 2024

Editor: Sandi Orlic

Reviewer(s): Disclosure of reviewer identity is with reference to reviewer comments included in decision letter(s). The following individuals involved in review of your submission have agreed to reveal their identity: Greta Reintjes (Reviewer #1); Jan Muschiol (Reviewer #2)

Transaction Report:

DOI: <https://doi.org/10.1128/spectrum.03036-23>

Re: Spectrum03036-23 (Decoupling between genetic potential and the metabolic regulation and expression in microbial organic matter cleavage across microbiomes)

Dear Dr. Zihao Zhao:

Thank you for the privilege of reviewing your work. Below you will find my comments, instructions from the Spectrum editorial office, and the reviewer comments.

Revision Guidelines

Sincerely,
Sandi Orlic
Editor
Microbiology Spectrum

Reviewer #1 (Comments for the Author):

Dear Authors,

In this manuscript, you have taken extensive omics data from the gut, marine, and soil environment and cross-compared the organic-matter degradation potential. The cross-habitat analysis shows detailed differences in the results, which is reflected well in the discussion. The comparison of the gene (potential) and protein (expressed) datasets yields intriguing results, which the

authors have discussed in the context of the conditions of each habitat.

I thank the authors for their efforts and the well-written manuscript. I have several suggestion points which I hope will help them to better structure and focus the manuscript. See attached document

Reviewer #2 (Comments for the Author):

The authors of the present manuscript analyzed and compared the CAZyme and peptidase profiles of each several hundred metagenomes, metatranscriptomes and metaproteomes from three different environments (marine, soil, human gut). Established bioinformatic methods were used to predict taxonomy, functionality, localization and the isoelectric point of the analyzed proteins. Elaborate statistical methods were used to identify links between these predicted parameters and the environments the samples were isolated from. In general, I enjoyed reading the paper, which in my opinion will be interesting for a broader readership, because such a high number of datasets were analyzed and more general conclusions were drawn. I have only a few comments, which I'd like to recommend the authors to consider to improve the manuscript.

1. l. 220-224: This sentence is misleading. AAs are not only acting on lignin. That's actually only a few families such as AA1 (laccases) and AA2 (peroxidases). The majority of the AA families (AA9-11, AA13-17) actually contain so called Lytic polysaccharide monooxygenases (LPMOs), which directly introduce chain breaks into recalcitrant polysaccharides such as chitin and cellulose to facilitate further breakdown by GHs. This needs to be considered especially since the authors showed that AAs in the marine environment are much more present compared to soil and human gut across all omic datasets analyzed (Fig. 2).
2. l. 241: I wonder how significant a value of $3.6 \pm 3.6\%$ is. However, I also need to admit that I am not an expert in statistics.
3. l. 301: Please change "cystine" to "cysteine" here and throughout the whole text. The term cystine actually refers to a disulfide bridge.
4. l. 475-477: The authors mention oxidative degradation of algal polysaccharides by AAs and refer to a reference for degradation of chitin. However, to the best of my knowledge chitin is not the most prominent polysaccharide in algae, although it might be present. Given the fact that many LPMOs (see comment 1.) are well known to degrade chitin or cellulose, AAs might indeed have an important role in degradation of marine algal polysaccharides beyond chitin and cellulose. Although this still needs to be verified.
- l. 509: Please specify the databases and their versions from which the datasets were downloaded. Alternatively, please specify the date of download.
- l. 576: Please specify the "specific packages" used in R.
- l. 850-860, Table 1: I wonder if BG is actually a proper measure/enzyme activity to look for in oceanic samples? Maybe not since cellulosic material is low abundant in marine carbohydrates compared to soil samples and polysaccharides are generally much more diverse and complex. Refer to for example Vidal-Melgosa et al. (<https://doi.org/10.1038/s41467-021-21009-6>).

Review of the manuscript entitled: Decoupling between genetic potential and the metabolic regulation and two expressions in microbial organic matter cleavage across microbiomes

In this manuscript, the authors have taken extensive omics data from the gut, marine, and soil environment and cross-compared the organic-matter degradation potential. The cross-habitat analysis shows detailed differences in the results, which is reflected well in the discussion. The comparison of the gene (potential) and protein (expressed) datasets yields intriguing results, which the authors have discussed in the context of the conditions of each habitat.

I thank the authors for their efforts and the well-written manuscript. I have several suggestion points which I hope will help them to better structure and focus the manuscript.

Introduction

Line 45-47, 50-52: Repetition.

Line 57-59: How does this affect enzyme production?

Line 63: *"Too large for the prokaryotic transporter system ."* Which transporter system? What about large transporters TonB or porins?

Line 72-73: Is it in the periplasm secretory? What access would periplasmic proteins have to the external environment if, as you stated, there are no transporters?

Line 82-83: Using multi-omics approaches from several studies or data from these studies??

Line 87: *"Microbially mediated substrate cleavage ."* Your analysis is only potential as you cannot say if the expressed proteins are active or the turnover rate. Do these data, then, in fact, yield cleavage results?

From the first sections of the introduction, it is not extremely clear what the aim is. Combine loads of omics data types to look at organic matter cleavage. Compare genes to proteins expressed. A precise, defined purpose would be great.

Line 103-106: *"Thus, we hypothesize that microbial cleavage mediated by extracellular enzymes revealed by different omics tools will exhibit distinct patterns between microbiomes and affect the role of microbes in element cycle"*. How do they affect the role of microbes in element cycles?? I don't understand this statement.

What about the extreme limitations in proteomics and resolution differences in the omics methods? With every proteomic data set, we have a huge proportion of hypothetical proteins for which we still don't know the function. This comment should not reflect poorly on your analysis; I just believe it is worth highlighting the potential deficit in omics, as experts in the field will directly think of this.

Line 102-103: This statement is false. There is a huge amount of data about how microbes respond both individually and as a community to organic matter supply.

Introduction general comment. I would be careful with statements of *"not well understood"*, *"unexplored"* and *"not well documented"*, in part your statement is incorrect. Focus on what we have and how your study potentially increases our knowledge.

Results & Discussion

Line 112: “leveraging” is A very unusual word in your context.

Line 113-117 Are the genes from the omics data or additional? If not how are these genes different from the datasets?

Line 146 – Rarefaction?

Line 147-149 Where is the secreted shown in Fig 1? How are figure 1 and S1 different? Is there complete repetition between these figures?

Line 176-177: Figure 1d does not show these parameters. Placement of reference is misleading.

Line 202-211: Figure 2 “*Functional and taxonomic composition*”. These figures are challenging to interpret. I find it nearly impossible to look at the details. From the text, I see you are referring to the general trends, and I would strongly suggest simply averaging the results per habitat and thereby reducing the plot complexity.

Furthermore, I would suggest showing all these repeated % results in a table. It would save you text and make the interpretation and overview easier. (Line 202-211 and Line 227-238).

Line 233: Is there any evidence that planctomycetes are better represented in proteomic data? Is it either from the identification side or from the protein extraction side? Do they have a higher protein content, large cell size??

Line 245-247 Nice

Figure S3-S5 These figures are equally difficult to interpret outside of more or less green etc. simplify.

Line 297-303. These details would be better interpreted from averaged graphs, which could even be added as a main figure if you would like to make the secreted/cytoplasmic comparison a key point.

Line 314-315: This is a really interesting finding that summarizes those complex graphics very well.

Line 321-322 How was the identification of the substrate done? By comparison to characterized proteins? A little detail here would be good as we have few characterized proteins.

Figure 3: It is very interesting to see that Algae carbohydrates are highly expressed in RNA in soil samples, do you have an explanation for this? Could it be methodological?

Line 343-345: This disclaimer should be mentioned earlier as a key point. Its sets the framework for the data interpretation.

Figure S8 very interesting figure worth placing as a main figure.

Line 374-377: very important point and well stated, however, no need to repeat it see Line 404

Paragraph line 378-412: This section could be shortened. I find that there is a high level of repetition and walk around. A short detailed description of your very interesting finds would be good.

Line 415-416: you didn't mention this section or its interest in the abstract/intro.

Line 424-429: Are these findings primarily related to coverage level?

Line 443-462: I find this section is a little misplaced and doesn't have a lot of data. I would suggest omitting it.

Line 466-468: how many hands? Suggest rephrasing your final findings.

Line 483-485: A clearer statement of findings is important.

Line 489- Universal uncoupling is a little board based on the limits in protein datasets.

Line 499 – repeat of decoupling statement.

Methods

Line 509: Which databases? Details for reproducibility are essential.

Line 530-532: Lack of detail. How was this done? Details please. How was the data reanalysed, what commands were use, what settings in the programs, what levels, were normalisations/standardisations done on the datasets??? What was used to calculate abundance gene coverage, total abundance, normalised abundance ?? Details please.

Additionally, did you do any normalisation to account for the difference in abundance/ coverage between proteomics and genomics.

Line 532-535: How were these keywords chosen? The major marine storage compound is laminarin and this is not in the list.

Figure legends:

Figure 1. a) is both a Venn diagram and rarefaction curve, this is not stated in the legend.

Reviewer #1 (Comments for the Author):

Review of the manuscript entitled: Decoupling between genetic potential and the metabolic regulation and two expressions in microbial organic matter cleavage across microbiomes

In this manuscript, the authors have taken extensive omics data from the gut, marine, and soil environment and cross-compared the organic-matter degradation potential. The cross-habitat analysis shows detailed differences in the results, which is reflected well in the discussion. The comparison of the gene (potential) and protein (expressed) datasets yields intriguing results, which the authors have discussed in the context of the conditions of each habitat.

I thank the authors for their efforts and the well-written manuscript. I have several suggestion points which I hope will help them to better structure and focus the manuscript.

We thank for the positive feedback and the detailed comments of the reviewer, which helped to improve the clarity of the manuscript.

Introduction

Line 45-47, 50-52: Repetition.

We removed the redundant sentence

Line 57-59: How does this affect enzyme production?

We delete this sentence

Line 63: "Too large for the prokaryotic transporter system ."Which transporter system? What about large transporters TonB or porins?

Here we mean all transporter systems. The TonB system can take substrate of around 1000 Da, but in this manuscripts, the carbohydrates and proteins are referred to as high molecular weight dissolved organic matter and particulate organic matter, which are >10,000 Da or visible particles. These substrates need to be hydrolyzed/degraded prior to assimilation

Line 72-73: Is it in the periplasm secretory? What access would periplasmic proteins have to the external environment if, as you stated, there are no transporters?

The cell-associated extracellular enzymes are bound to the cell in the periplasmic space/S-layers with the cleavage site exposed to the ambient environment. The cleavage site can 'catch' large polymers and hydrolyze them. The hydrolysate will be then assimilated by transporters close to the cell-associated extracellular enzymes. In Bacteroidetes, such structure is defined as polysaccharide utilization loci (PULs), and similar structures can also be found in other microbes like Gammaproteobacteria. We use the figure from Reintjes et.al., 2018 to show how cell-associated extracellular enzymes are related to substrate cleavage.

Fig. 1 Schematic diagram of three main mechanisms of HMW substrate utilisation. Selfish: cells use surface associated enzymes to bind and partially degrade polysaccharides, which are directly taken up into the periplasm for further degradation with little to no production of extracellular hydrolysis products. Sharing: cells use surface-associated

or 'free' extracellular enzymes to degrade polysaccharide to sizes suitable for uptake. Causes production of extracellular hydrolysis products (public goods). Scavengers: cells do not or cannot produce enzymes for the hydrolysis of polysaccharides, but take up the hydrolysis products produced by other organisms

Line 82-83: Using multi-omics approaches from several studies or data from these studies??

These are data from several studies, we corrected this in the text (line 77)

Line 87: *“Microbially mediated substrate cleavage .”*Your analysis is only potential as you cannot say if the expressed proteins are active or the turnover rate. Do these data, then, in fact, yield cleavage results?

We changed it to ‘molecular basis of microbially mediated substrate cleavage among contrasting biomes.’(line 84)

From the first sections of the introduction, it is not extremely clear what the aim is. Combine loads of omics data types to look at organic matter cleavage. Compare genes to proteins expressed. A precise, defined purpose would be great.

Line 73-77: An additional sentence has been added to highlight the aim and purpose of this manuscript.

Line 103-106: *“Thus, we hypothesize that microbial cleavage mediated by extracellular enzymes revealed by different omics tools will exhibit distinct patterns between microbiomes and affect the role of microbes in element cycle”*. How do they affect the role of microbes in element cycles?? I don’t understand this statement.

We delete the final part of the sentence: *“and affect the role of microbes in element cycle”* has been deleted

What about the extreme limitations in proteomics and resolution differences in the omics methods? With every proteomic data set, we have a huge proportion of hypothetical proteins for which we still don’t know the function. This comment should not reflect poorly on your analysis; I just believe it is worth highlighting the potential deficit in omics, as experts in the field will directly think of this.

We address the limited resolution of metaproteomic analysis in line 80-83: *“Particularly, proteins recovered from the metaproteome reflect the major functions mediated by microbes, although the resolution of metaproteomic analysis is relatively low compared to metagenomics and metatranscriptomics.”*

Regarding the hypothetical proteins, in the marine metaproteome, around 30% proteins in the metaproteomic analysis are hypothetical (Cohen et. al., 2021, Zhao et. al., 2023), this is much lower than the metagenome and metatranscriptome, where >50% of the predicted genes are hypothetical (Sunagawa et. al., 2015, Carradec et. al., 2018). Also, we used the genes from the metagenome/metatranscriptome as the database for metaproteomic search. If the corresponding gene in the database is hypothetical, the protein (if also detected in the metaproteome) will also be hypothetical. All the CAZymes and peptidases detected in the metaproteome are all from genes annotated as CAZymes or peptidases.

Line 102-103: This statement is false. There is a huge amount of data about how microbes respond both individually and as a community to organic matter supply.

We delete this sentence and changed it to : *‘This taxonomic change will further affect the patterns of microbial degradation on organic matter in their habitats.’* (line 101-103)

Introduction general comment. I would be careful with statements of *“not well understood”*, *“unexplored”* and *“not well documented”*, in part your statement is incorrect. Focus on what we have and how your study potentially increases our knowledge.

We deleted these phrases

Results & Discussion

Line 112: *“leveraging”* is A very unusual word in your context.

Changed to ‘using’ (line 114)

Line 113-117 Are the genes from the omics data or additional? If not how are these genes different from the datasets?

These genes are from the same omics dataset as reported in the corresponding publications

Line 146 – Rarefaction?

Corrected

Line 147-149 Where is the secreted shown in Fig 1? How are figure 1 and S1 different? Is there complete repetition between these figures?

Fig S1 is not a repetition of Fig 1, it shows the profile of cytoplasmic CAZymes and peptidases, although the patterns are quite similar to secretory ones.

Line 176-177: Figure 1d does not show these parameters. Placement of reference is misleading.

We removed the figure reference here.

Line 202-211: Figure 2 “*Functional and taxonomic composition*”. These figures are challenging to interpret. I find it nearly impossible to look at the details. From the text, I see you are referring to the general trends, and I would strongly suggest simply averaging the results per habitat and thereby reducing the plot complexity.

We revised Fig2 by only showing the major contributors and group the minor taxa into ‘others’ to highlight the main findings.

Furthermore, I would suggest showing all these repeated % results in a table. It would save you text and make the interpretation and overview easier. (Line 202-211 and Line 227-238).

We tried to make a table with mean+SD, but we ended up with 6 columns (2 enzymes *3 omics) and 15 rows (3 microbiome * 5 major taxa). The detailed % of the taxonomy can be found in supplementary dataset 4.

Line 233: Is there any evidence that planctomycetes are better represented in proteomic data? Is it either from the identification side or from the protein extraction side? Do they have a higher protein content, large cell size??

We don’t have such data for soil Planctomycetes, but in the marine metaproteome, we found a similar pattern for Nitrospinae, which has a low cell abundance but a high expression of nitrite oxidoreductase due to their large cell (Zhao et. al., 2023). Also in marine environments, Gammaproteobacteria are the major contributors to secretory CAZymes and peptidases, but their cell number is low and cell size is large. Thus, soil Planctomycetes may also have large cells.

Line 245-247 Nice

Thanks!

Figure S3-S5 These figures are equally difficult to interpret outside of more or less green etc. simplify.

We revised these figures by highlighting the major groups.

Line 297-303. These details would be better interpreted from averaged graphs, which could even be added as a main figure if you would like to make the secreted/cytoplasmic comparison a key point.

We added fig S6 for comparison as suggested

Line 314-315: This is a really interesting finding that summarizes those complex graphics very well.

Thanks!

Line 321-322 How was the identification of the substrate done? By comparison to characterized proteins? A little detail here would be good as we have few characterized proteins.

In the CAZyme sequence database dbcan, there is the map file between CAZyme families and

substrate: <https://bcb.unl.edu/dbCAN2/download/Databases/fam-substrate-mapping-08252022.tsv>

We also added this to the method part (line 475)

Figure 3: It is very interesting to see that Algae carbohydrates are highly expressed in RNA in soil samples, do you have an explanation for this? Could it be methodological?

Yes, this is because there is a functional overlap between CAZyme families targeting different types carbohydrate (GH16 and GH55). For example, GH55 contains β -1,3-glucanases (endo and exo types) which are widely used by both marine and soil microbes, also, GH55 contains laminarinase ((endo-1,3(4)- β -Glucanase) which targets on the carbohydrates of algae (laminarin). Similarly, GH16 active on β -1,4 or β -1,3 glycosidic bonds in various glucans and galactans can hydrolyze both lichenin in soil ecosystem and carrageenan in marine environment.

We added discussions to address this limitation (line 335-339) : “This high level of gene/transcript abundance of CAZymes targeting algal carbohydrates in the soil microbiome is because the GH55 and GH16 are also widely used by soil microbes hydrolyzing glycosidic bonds in various glucans, and laminarin and carrageenan in marine environment are also rich in β -1,4 or β -1,3 glycosidic bonds.”

Line 343-345: This disclaimer should be mentioned earlier as a key point. It sets the framework for the data interpretation.

We address the limited resolution of metaproteomics in the introduction (line 80-83): “Particularly, proteins recovered from the metaproteome reflect the major functions mediated by microbes, although the resolution of metaproteomic analysis is relatively low compared to metagenomics and metatranscriptomics.”

Figure S8 very interesting figure worth placing as a main figure.

We move Fig S8 to Fig.3C

Line 374-377: very important point and well stated, however, no need to repeat it see Line 404

We deleted the redundant sentence

Paragraph line 378-412: This section could be shortened. I find that there is a high level of repetition and walk around. A short detailed description of your very interesting finds would be good.

We reduced the text in this section substantially to avoid repetition

Line 415-416: you didn't mention this section or its interest in the abstract/intro.

We added in the abstract: “Such variations lead to decoupled relationships between CAZymes and peptidases from genetic potentials to protein expressions due to the different response of key players towards organic matter sources and concentrations.” (line 22-25)

Also in the introduction (line 91-93): “the differences in C:N stoichiometry between habitats might lead to distinct genetic features in microbial communities and the expression level of corresponding enzymes might display contrasting patterns due to the spontaneous response of the microbial community.”

Line 424-429: Are these findings primarily related to coverage level?

Yes.

Line 443-462: I find this section is a little misplaced and doesn't have a lot of data. I would suggest omitting it.

We deleted this section

Line 466-468: how many hands? Suggest rephrasing your final findings.

We deleted the entire section.

Line 483-485: A clearer statement of findings is important.

We changed it to “Our results reveal that the key taxonomic microbial groups in organic matter hydrolysis living in different habitats can be differentiated by omics tool and the changes in enzyme production between different taxa lead to a decoupling between the genomic potential and corresponding transcripts/proteins. Key players expressing a larger fraction of the enzymes than expected based on their genomic potential cause this uncoupling between genetic potential and enzyme expression. These key players probably act as the driving force in the element cycling. Thus, studies based exclusively on metagenomic analyses to elucidate functions in microbial communities do not reflect the actual distribution patterns of ectoenzyme secretion.” (line 436-439)

Line 489- Universal uncoupling is a little board based on the limits in protein datasets.

We delete ‘universal’

Line 499 – repeat of decoupling statement

We delete this sentence

Methods

Line 509: Which databases? Details for reproducibility are essential.

The version/ download date was added to the text (line 456&458)

Line 530-532: Lack of detail. How was this done? Details please. How was the data reanalysed, what commands were use, what settings in the programs, what levels, were normalisations/standardisations done on the datasets??? What was used to calculate abundance gene coverage, total abundance, normalised abundance ?? Details please.

Additionally, did you do any normalisation to account for the difference in abundance/ coverage between proteomics and genomics.

We add details how we evaluated the relative abundance of CAZymes targeting different types of carbohydrates. The gene/transcript/protein abundances of corresponding CAZymes from omics dataset were normalized to the total secretory CAZymes, thus, it is all relative abundance (%)

Now line 482-485: “To evaluated the relative abundance of CAZymes targeting specific carbohydrates in the secretory CAZyme pool in omics datasets, the gene/transcript/protein abundances of CAZymes affiliated to families listed in Table S3 were summed and normalized to all secretory CAZymes”

Line 532-535: How were these keywords chosen? The major marine storage compound is laminarin and this is not in the list.

We thank the reviewer pointing out this, we did search for laminarin, but forgot to add to the text. From Table S3 there are quite some CAZyme families targeting laminarin (i.e. GH55, GH64 and GH158), this is also why the transcripts of CAZymes targeting algal carbohydrate exhibited high relative abundances in the soil metatranscriptome, due to GH55, a β -1,3-glucanases, widely used by both marine and soil microbes. Now we have corrected this mistake.

Figure legends:

Figure 1. a) is both a Venn diagram and rarefaction curve, this is not stated in the legend.

We correct the legend, thanks.

Reviewer #2 (Comments for the Author):

The authors of the present manuscript analyzed and compared the CAZyme and peptidase profiles of each several hundred metagenomes, metatranscriptomes and metaproteomes from three different environments (marine, soil, human gut). Established bioinformatic methods were used to predict taxonomy, functionality, localization and the isoelectric point of the analyzed proteins. Elaborate statistical methods were used to identify links between these predicted parameters and the environments the samples were isolated from. In general, I enjoyed reading the paper, which in my opinion will be interesting for a broader readership, because such a high number of datasets were analyzed and more general conclusions were drawn. I have only a few comments, which I'd like to recommend the authors to consider to improve the manuscript.

We thank reviewer for the positive comments on our manuscripts. We also appreciated the comments on lytic polysaccharide monoxygenases related CAZymes. We incorporated the suggestion in the revised version, our answers to the comments can be found below as well as in the revised manuscript.

1. l. 220-224: This sentence is misleading. AAs are not only acting on lignin. That's actually only a few families such as AA1 (laccases) and AA2 (peroxidases). The majority of the AA families (AA9-11, AA13-17) actually contain so called Lytic polysaccharide monoxygenases (LPMOs), which directly introduce chain breaks into recalcitrant polysaccharides such as chitin and cellulose to facilitate further breakdown by GHs. This needs to be considered especially since the authors showed that AAs in the marine environment are much more present compared to soil and human gut across all omic datasets analyzed (Fig. 2).

We thank the reviewer for pointing this out. We add discussion to address the role of LPMOs in line 219-226. Now it reads: "CAZymes with AAs are also known as lytic polysaccharide monoxygenases [LPMOs] (33) involved in polysaccharide degradations because crystalline polysaccharides in plant and algae are less accessible to degradative CAZymes (like glycoside hydrolases [GHs], carbohydrate esterase [CEs] and polysaccharide lyases [PLs]) (34, 35). By oxidizing the surface of polysaccharides crystal structure and disrupting the polysaccharides topology, the oxidative activity of AAs create tractable chain ends for further depolymerization mediated by hydrolytic CAZymes like GHs (36)."

2. l. 241: I wonder how significant a value of $3.6 \pm 3.6\%$ is. However, I also need to admit that I am not an expert in statistics.

Here the $3.6 \pm 3.6\%$ shows the relative abundance of genes encoding Planctomycetes-affiliated secretory CAZyme in the secretory CAZyme gene pool of soil microbiome. We compared this value to the relative protein abundance of Planctomycetes-affiliated secretory CAZymes in the soil metaproteome, where Planctomycetes-affiliated secretory CAZymes contributed $45.6 \pm 21.5\%$ to the secretory CAZyme pool, which was significantly higher than the relative abundance of corresponding genes. We rephrased this sentence as follows to clarify.

Now line 246-249: "This low representation of Planctomycetes is in striking contrast to their major role in secretory CAZyme production in the soil metaproteome (Fig. 2b, supplementary dataset 4), where Planctomycetes-affiliated secretory CAZymes contributed $45.6 \pm 21.5\%$ to the secretory CAZyme pool and were significantly higher than the relative abundance of the corresponding genes (Wilcoxon test, $p < 0.05$)."

3. l. 301: Please change "cystine" to "cysteine" here and throughout the whole text. The term cystine actually refers to a disulfide bridge.

Corrected as suggested

4. l. 475-477: The authors mention oxidative degradation of algal polysaccharides by AAs and refer to a reference for degradation of chitin. However, to the best of my knowledge chitin is not the most prominent polysaccharide in algae, although it might be present. Given the fact that many LPMOs (see comment 1.) are well known to degrade chitin or cellulose, AAs might indeed have an important role in degradation of marine algal polysaccharides beyond chitin and cellulose. Although this still needs to be verified.

We deleted this section to avoid misleading statements, which has been also suggested by reviewer #1

l. 509: Please specify the databases and their versions from which the datasets were downloaded.

Alternatively, please specify the date of download.

The version and download date were added to the text (line 456&458)

l. 576: Please specify the "specific packages" used in R.

We rephrased as following to show how R packages were used for the analysis: “Statistics and visualization were performed in R (<https://www.r-project.org/>). Vegan, rtk and wilcox.test were used for ordination, diversity calculation and significance test, ggplot2 were used for data visualization.” (line 524-526)

I. 850-860, Table 1: I wonder if BG is actually a proper measure/enzyme activity to look for in oceanic samples? Maybe not since cellulosic material is low abundant in marine carbohydrates compared to soil samples and polysaccharides are generally much more diverse and complex. Refer to for example Vidal-Melgosa et al. (<https://doi.org/10.1038/s41467-021-21009-6>).

We totally agree with this comment and this is one of the reasons we conducted such ‘multi-omics’ comparison between distinct microbiomes. Current enzymatic activity measurements in marine environment still use BG as a proxy for betaglucosidically linked carbohydrate cleavage measurements. As mentioned above, AAs may play an important role in both chitin and algal polysaccharides degradation in marine environment but might be overlooked.

However, the substrate analogs for AAs enzymatic activity measurement need to be tested with further focus on the specific types of AAs secreted by marine microbes and the omics analysis might provide molecular basis for the selection of such analogs.

In the revised manuscript, we deleted this section and table 1 because the link from omics results only showed very limited link to the enzymatic activity measurement.

Re: Spectrum03036-23R1 (Decoupling between genetic potential and the metabolic regulation and expression in microbial organic matter cleavage across microbiomes)

Dear Dr. Zihao Zhao:

Your manuscript has been accepted, and I am forwarding it to the ASM production staff for publication. Your paper will first be checked to make sure all elements meet the technical requirements. ASM staff will contact you if anything needs to be revised before copyediting and production can begin. Otherwise, you will be notified when your proofs are ready to be viewed.

Sincerely,
Sandi Orlic
Editor
Microbiology Spectrum

Reviewer #1 (Comments for the Author):

Thank you for your review responses and for considering my suggestions. Best of luck in your future endeavors.